# The Effect of Nordic Walking Training with Poles with an Integrated Resistance Shock Absorber on the Functional Fitness of Women over the Age of 60

**DOI:** 10.3390/ijerph17072197

**Published:** 2020-03-25

**Authors:** Katarzyna Marciniak, Janusz Maciaszek, Magdalena Cyma-Wejchenig, Robert Szeklicki, Zuzanna Maćkowiak, Dorota Sadowska, Rafał Stemplewski

**Affiliations:** 1Poznan University of Physical Education, 61-871 Poznan, Poland; katarzyna.anna.m@gmail.com (K.M.); magdalenacyma@gmail.com (M.C.-W.); szeklicki@awf.poznan.pl (R.S.); zuzannamackowiak84@gmail.com (Z.M.); stemplewski@awf.poznan.pl (R.S.); 2National Research Institute, 03-301 Warsaw, Poland; sadowska.dorota@hotmail.com

**Keywords:** physical activity, Nordic walking, functional fitness, Senior Fitness Test, functional training, aging

## Abstract

Inadequate levels of physical activity among older people lead to a gradual decline in self-reliance and consequent dependence on other people. The aim of the study was to evaluate the impact of Nordic walking training with poles with an integrated resistance shock absorber on the functional fitness of older women. Forty-two women (*M*_age_ = 64.7 ± 3.15 years) were randomly assigned into the experimental group—training with poles with an integrated resistance shock absorber, EG (*n* = 21) and the control active group—training with classic poles, CG (*n* = 21). Functional fitness was measured with the Senior Fitness Test before and after an intervention lasting for 8 weeks (2 training sessions × 75 minutes per week). Two-way ANOVA revealed statistically significant interaction effects for aerobic endurance (*F* = 14.47, *p* < 0.001) and upper body strength (*F* = 5.98, *p* < 0.05), indicating greater improvement in the experimental group. Nordic walking training both with classic poles and with poles with an integrated resistance shock absorber is beneficial for older people and improves functional fitness over a short time period. However, the poles with an integrated resistance shock absorber provide additional resistance effort during marching, which causes increased muscle activation and results in improved muscle strength and aerobic endurance. Based on these results, it can be concluded that this kind of training could be applied in the complex health programs of seniors.

## 1. Introduction

One of the basic activities of everyday life is walking, which might be crucial in delaying decline in physical fitness and prevent limitation in self-reliance and consequent dependence on other people [1,2,3,4,5,6]. Nordic walking (NW) is one of the physical activities that can positively influence the walking ability. Proper physiological gait requires adequate body coordination and the synchronization of individual muscle groups. Gait quality depends on the proper range of mobility in all joints of the lower limbs as well as the muscle strength necessary to ensure optimum control over joint movements. Muscle strength decreases with age due to a decrease in the size and number of muscle fibers as a result of sarcopenia [7,8].

Considering the impact of sarcopenia-related changes on functional fitness, marching is recommended for older people as a form of prevention and rehabilitation [9,10]. The open question remains whether such a stimulus is sufficient. Sánchez-Sánchez et al. [11] noted the unclear impact of walking on the health of older people. However, other studies have shown that endurance exercises can even attenuate sarcopenia [12,13]. Martins et al. [14] compared the effects of strength training and aerobic training, confirming the positive effect of both types of training on the functional fitness of older people.

According to Sherrington et al. [15], a balance exercise program for seniors demonstrates an improved response compared to that of aerobic exercise. Clemson et al. [16] in turn confirmed the effect of strength and balance exercises on the improvement of the indicators related to the dynamic balance of the body. The research by Ramsbottom et al. [17], which incorporated twenty-four weeks of strength training, also showed increased muscle strength in the lower limbs and dynamic body balance. Additionally, Martone et al. [18] showed the effectiveness of strength training in the maintenance of adequate body mass and strength among seniors.

Given this information, it is important to determine the type of training that would include all critical elements of physical fitness and thereby influence the maintenance of fitness levels or slow down aging processes. It is equally important that physical activities could promote social contact between people. NW is a popular type of physical activity among older people. The use of walking poles causes additional muscle involvement during walking [19]. The results of Lee and Park [20] confirmed a greater improvement in body balance and leg strength in the NW training group compared to that in the group doing general exercises.

Additionally, Parkatti et al. [21] concluded that NW training significantly influenced the functional fitness of older people. They reported that the application of 60 minutes of NW training conducted over a 9-week period results in statistically significant improvement in the strength of the lower and upper limbs and the flexibility of the body among older people, as measured by the Senior Fitness Test (SFT).

The new form of NW is training with modified poles, which allows combining aerobic and strength training. These modified poles contain a built-in resistance shock absorber (RSA). An elastic tape between two permanent elements in RSA poles allows additional resistance to be obtained by increasing the overall intensity of exercise, actual VO_2_, and calorie consumption. One can expect that the effect of this kind of activity could be similar to those obtained during training with elastic bands, which is a popular form of training. Oesen et al. [22] showed that low-intensity resistance exercises using elastic bands are safe and beneficial in improving the functional performance of institutionalized seniors. Usually, in combined training, endurance and resistance exercises are provided alternately. It is a question of whether the effect of training with RSA poles will be different, taking into account that elements of resistance training are simultaneously performed with endurance activity. It might be necessary to take into account potential interference phenomena that occur in concurrent training and may lead to decreased strength gain. However, this interference does not impair cardiovascular adaptation [23].

According to the authors’ best knowledge, no research has been published thus far on the effect of NW training with RSA poles on functional fitness in seniors.

The purpose of the study was to evaluate the effect of NW with RSA poles on the functional fitness of women over the age of 60 years. We hypothesized that training with RSA poles is more beneficial than Nordic walking with classic poles for the functional fitness of older women in terms of the strength of the upper and lower limb muscles, aerobic endurance, and agility.

## 2. Methods

### 2.1. Participants

The study included 42 women aged over 60 years (*M*_age_ = 64.7 years, age range: 60–71 years). All women were informed in detail of the study and gave their written consent to the experimental procedure. Before the implementation of the research project, the local Bioethical Committee granted its approval (no. 421/16).

The recruitment was carried out using ads in social media such as the local newspapers and the Internet. During the interview, the questions regarding the following were asked: age; type of medication taken, illness, and lifestyle.

The following inclusion criteria were taken into account: age above 60; no medical restrictions on participation; no experience in marching training or other regular sports activities (more than once a week for more than 20 minutes with moderate–vigorous intensity, e.g., jogging, cycling, swimming, etc.). Subjects with the presence of at least one of the factors: musculoskeletal disorders that prevent independent movement, dizziness, diabetes type 2, obesity, and use of blood pressure-lowering drugs were excluded. Before the research program was implemented, the women provided a primary care physician’s certificate of no contraindications to moderate exercise.

Forty-seven women were randomly (Excel software) assigned to the study into two groups:

Experimental group (EG)—Nordic walking with RSA poles—initial *n* = 23;

Control group (CG)—Nordic walking with classic poles—initial *n* = 24.

Before the study, 2 women in the experimental group and 3 women in the control group resigned from the trial without a reason, resulting in 21 women in both groups (Figure 1).

None of the subjects had experience in NW with RSA or classic poles. All people were familiarized with the planned, detailed course of study, schedule, and program of classes. The informative material also contained an outline of the exercises that should be implemented during training sessions. Each exercise was thoroughly described and illustrated by an image (printed summary with photos). The subjects’ characteristics of both groups are shown in Table 1.

### 2.2. Procedure

A general overview of the experiment is shown in Figure 2.

**Intervention.** Both groups trained at the same time, twice a week for 8 weeks, for a total of 16 training sessions. Women from the CG used classic NW poles (Nordic Walking Race 80% Carbon, Fizan, Italy), while the experimental group used poles with a built-in resistance shock absorber—elastic resistance of 4 kg (Slimline Bungy Pump, Sports Progress International AB, Sweden). 

Each training session began with 10–15 minutes of warm-up, during which all participants performed vigorous body movements, such as sweeping legs, and arms, climbing feet, bending knees, and elbows. After half of the planned distance of walking (about 1.7–2.2 km with a speed of about 6 km/h), participants did strength exercises and balance training (15 minutes). Strength exercises were based on overcoming the elastic resistance of the shock absorber of the RSA sticks in various positions of the body. Balance exercises were performed in the conditions of a reduced base of support, for example, standing on one leg. After going the rest of the distance, stretching exercises took place (15 minutes). During the whole intervention, the distance and number of exercises performed were gradually increased (from 3.5 to 4.5 km while walking; from 8 repetitions to 12 repetitions for exercises, respectively). The training plan included American College of Sports Medicine (ACSM) recommendations for adults and healthy older people: 1 set of 8–10 exercises for major muscle groups at least 2 days a week with 8–12 repetitions for each exercise [1].

Classes were held in accordance with the principles of Nordic walking training, and the trainer (K.M.) had the appropriate qualifications of instructor and trainer (International Nordic Walking Association). Participants were familiarized with the technique of marching and the use of equipment during an additional 60-minute instructional session before starting the experiment. All participants were instructed to achieve an exercise intensity corresponding to heart rate 100–120 bpm. The participant was supposed to walk as fast as possible but at speeds that still allowed them to speak. Before each session, five randomly selected women received a heart rate monitor (Polar monitor) to control the intensity of the exercise. The training took place in a city park. Subjects were walking along the park’s inner lanes. The length of the gap was measured using the Endomondo application. 

A minimum attendance of 80% was adopted, which required participation in at least 13 training sessions.

**The technique of marching with RSA poles.** The marching technique with RSA poles is similar to that of classic NW. In both cases, the body’s movement is compatible with the individual phases of the gait, but limbs work with more power and in a wider range of motion than that in free walking. Unlike the NW poles, RSA poles do not give the exercising person stable support. Fitted against the facing resistance, they bend in contact with the ground. The rhythmic, alternate movement of the limbs and the repulsion of the poles from the ground increase the speed of the walk and require proper coordination from the exercising person. The stance of the walking pole must be oblique to the rear (approximately 45°). As a result of acceleration, the body is tilted forward with the center of gravity being shifted to the body and a noticeable increase in the workings of the limbs and trunk. Release of the pressure from the RSA might cause a sensation of changes in body balance.

**Technical description of RSA poles.** The Slimline 4, which was used for the intervention, has a trekking handle with a strap that can attach to the hands. The bottom of the pole comprises two parts. One of these parts hides inside the pole, and the other part is used to adjust the length. The adjustment mechanism is simple. The pole lock is made by tightening the top of the screw with the plastic cap. To adjust the length of the poles, consider the length of the shock absorber and add approximately 20 cm more than the NW poles. The correct height of the RSA poles is set midway between the points defined by the nipples and armpit. The construction of the RSA poles slightly modifies body position during walking in comparison to that of classic NW (Figure 3).

**Measurement of functional fitness.** There was blinding of all assessors who measured functional fitness. The main researcher (K.M.) and the researcher calculating results (R.S.) were excluded from measurements. Functional fitness was measured before (D.S., J.M.) and after (M.C.-W., Z.M.) the experiment (pre, post) with use of the SFT. Both phases of the study (pre, post) were completed at a similar time during the morning hours. The SFT, which was used in this study, has been shown to have content and construct validity as well as good test–retest reliability [24,25,26].

Four tests from the SFT were performed to evaluate these performance measures that are necessary to maintain independence and safe daily activities:A.The 30-second chair stand test: assessment of the lower body strength needed for walking, climbing stairs, getting up from a chair, etc. The result of the test is the number of full stand-ups from the chair within 30 seconds with upper limbs crossed on the chest.B.Arm curl test: assessment of upper body strength needed for activities requiring lifting or moving of objects. The result of this test is the number of full bends in the elbow in 30 seconds with a weight of 2.27 kg.C.The 2-minute step test: estimation of aerobic capacity. The result was the number of full steps taken in place in 2 minutes—the number of right knee repetitions to the required height.D.The 8-foot up and go test: agility score/dynamic balance level in tasks requiring fast maneuvers such as getting in and out of public transport.

Tests were used strictly according to the original recommendations of Rikli and Jones [24].

### 2.3. Statistics

Statistical analyses were computed using Statistica v. 13.0 software (TIBCO Software Inc., Palo Alto, CA, USA). Statistical significance was defined as *p* ≤ 0.05. The main calculations for assessing the variance of dependent variables were based on two-way ANOVA (*F*-test) analysis methods. An analysis was made with repeated measurements before and after training (“time” factor with two levels—pre and post) with two levels of an intergroup factor (“group”—EG and CG) being taken into account. For interaction effects (“time × group”) and main effects (“time” and “group”), the eta-squared effect size was calculated. The effect size indicates the percent of variance explained by particular effects of the dependent variable. To compare the average values of dependent variables (pre–post within each group), Scheffe detailed post hoc comparisons were conducted. There was also a calculated percentage of differences between pre- and post-training.

## 3. Results

A statistically significant time effect for all parameters of functional fitness was observed: 30-second chair stand (*F* = 14.42, *p* < 0.001, *ɳ^2^* = 0.27), arm curl (*F* = 44.20, *p* < 0.001, *ɳ^2^* = 0.52), 2-minute step (*F* = 26.54, *p* < 0.001, *ɳ^2^* = 0.40), and 8-foot up and go (*F* = 88.27, *p* < 0.001, *ɳ^2^* = 0.69). In the 30-second chair stand test (Figure 4A), there was an increase in post-training values of 20% for the EG (Scheffe post hoc *p* < 0.01). In the 8-foot up and go test (Figure 4D), a decrease of 16% and 12%, respectively, for in EG and CG, was found. In both cases, post-training values were significantly lower than initial values (*p* < 0.001).

At the same time, the interaction effect was observed for the 2-minute step test (*F* = 14.47, *p* < 0.001, *ɳ^2^* = 0.27). An increase in post-training values of 21% in the EG (*p* < 0.001) was observed (Figure 4C). A significant interaction effect was also noticed for strength and endurance of the upper body in the arm curl test (*F* = 5.98, *p* < 0.05, *ɳ^2^* = 0.13), with significant post-training changes in the EG (*p* < 0.001) and the CG (*p* < 0.05). Post-training values were higher than pre-training values by 24% and 12%, respectively (Figure 4B).

## 4. Discussion

Exercises for older people must be simple and easy to imitate and should not involve a high financial cost if the goal is to maintain continuous participation in the classes. Marching with RSA poles combines elements of an aerobic workout with resistance training which is under ACSM recommendations [1]. In addition, this form of physical activity can be conducted in groups, which is important for the development of social contacts. According to the recommendations of the ACSM, a training plan for healthy older people should include aerobic exercises that influence endurance, resistance, and exercise to improve flexibility.

The most important effect of the current experiment is an improvement in the evaluated variables related to functional fitness in a short time period (8 weeks; only 16 training sessions). However, NW with RSA poles seems to provide a greater overall benefit. As a result of the experiment, statistically significant interaction effects were observed in the case of a 2-minute step test and arm curl test, indicating a greater improvement in the EG. The effects of the experiment confirm the improvement of performance parameters that are considered important for seniors’ health, i.e., aerobic endurance and muscle strength.

Marching with RSA poles is a new form of the popular activity of Nordic walking, and comparable data are limited. The obtained results might be compared to those of other studies involving training with aerobic and resistance exercises among older people. As observed in this research, the main time effects of marching training for both the EG and the CG on strength, endurance and agility are similar to those of other studies in which mixed exercise programs were employed [10,25,27,28,29].

As noted above, in the actual study, a higher improvement in endurance measured with the use of a 2-minute step test was observed in participants marching with RSA poles. Generally, other research has shown a positive effect of marching training on aerobic capacity [19,30]. However, a more recent study by Ozaki et al. [31] showed the crucial role of the intensity of walking in combined training. The authors suggested that the overall intensity of training should reach an effective level (50% VO_2_ max) for aerobic capacity improvement [32], which might explain the results obtained in our study. Although both groups (EG and CG) participated in the same exercise program and covered the planned distance at the same time, participants marching with RSA poles had to do additional work, each time overcoming the elastic resistance of 4 kg. The walking speed (and consequently the distance) of the participants could have been slightly different corresponding to the adopted intensity level.

In the analysis of the obtained results, it should be taken into account that characteristics of gait with the use of various NW poles (as well as without poles) might be different as in case of gait realized on a different surface. Shi et al. [33] observed differences in the structure of movement during over-ground walking and treadmill walking. They found shorter stride length, less stride time, and a reduced correlation between gait and upper trunk features on a treadmill. A similar situation might have occurred in our study. It is an open question whether there is a difference in the kinematics of gait with or without NW poles as well as whether applied training influenced on biomechanical characteristics of gait realized in everyday life. Shi et al. [33] suggested that changes in gait in nonstandard conditions might be related to fall risk prevention, which is very important among older people. Undoubtedly, it is an important direction for future studies.

Improved strength is usually reported as an effect of resistance training [34]. However, the results of research related to the effect of marching training on muscle strength are not consistent. Takeshima et al. [19] showed a positive effect of NW and resistance exercises and no effect for marching without poles. Similarly, Lee and Park [20] compared the effect of NW training with generalized exercises, which resulted in a greater improvement in lower body muscle strength in the NW group. There was no significant change in the strength of the upper body muscles. On the other hand, Song et al. [34,35] observed a similar positive effect on upper limb strength when comparing NW and resistance training. The current study has confirmed the hypothesis of a greater impact of NW with RSA poles on upper body strength, which may be related to the need to work with upper-extremity resistance. When individuals feel the pressure of the cradle shock, concentric work of the muscles of the brachial extensors is performed. On the other hand, there were no differences in training effects between groups on the strength of the lower body and agility, possibly due to the similar training load of the lower body parts for both types of training as well as the same type of coordination exercises used. It is also possible that the interference effect occurred, and the potential increase of muscle strength was lowered.

## 5. Limitations

This study has several limitations. First, all the participants were females in the current study. From this perspective, research on the participation of men is needed. Second, there was no comparison to strength training alone, which might be useful for the exclusion of potential interference effects. Third, not all environmental factors were controlled, such as the quality of food, caloric intake, lifestyle, and other activities. On the other hand, to lower the potential Hawthorne effect, an active control group was employed. Next, the differences in biomechanical characteristics of gait with NW poles and RSA poles and potential changes in gait structure as the effect of applied training were not investigated. This would make the analysis of results obtained in functional tests clearer as well as indicate possible benefits in the context of fall risk decrease.

## 6. Conclusions

In the experiment, significant improvement in functional fitness was observed in both groups after only 16 training sessions. However, to the best of our knowledge, this is the first time that NW with RSA poles, which combines classic NW and resistance exercises, seems to provide greater overall benefits in endurance and upper body strength. Participants marching with RSA poles had to do additional work with their upper extremities, thus increasing the overall exercise intensity in comparison to that of classic NW. Moreover, NW with RSA poles probably makes it easier to reach the physiologically effective level of exercise intensity in training among elderly women.

NW with RSA poles should be considered an engaging form of physical activity that is beneficial and fulfills ACSM criteria for exercise among elderly women.

## Figures and Tables

**Figure 1 ijerph-17-02197-f001:**
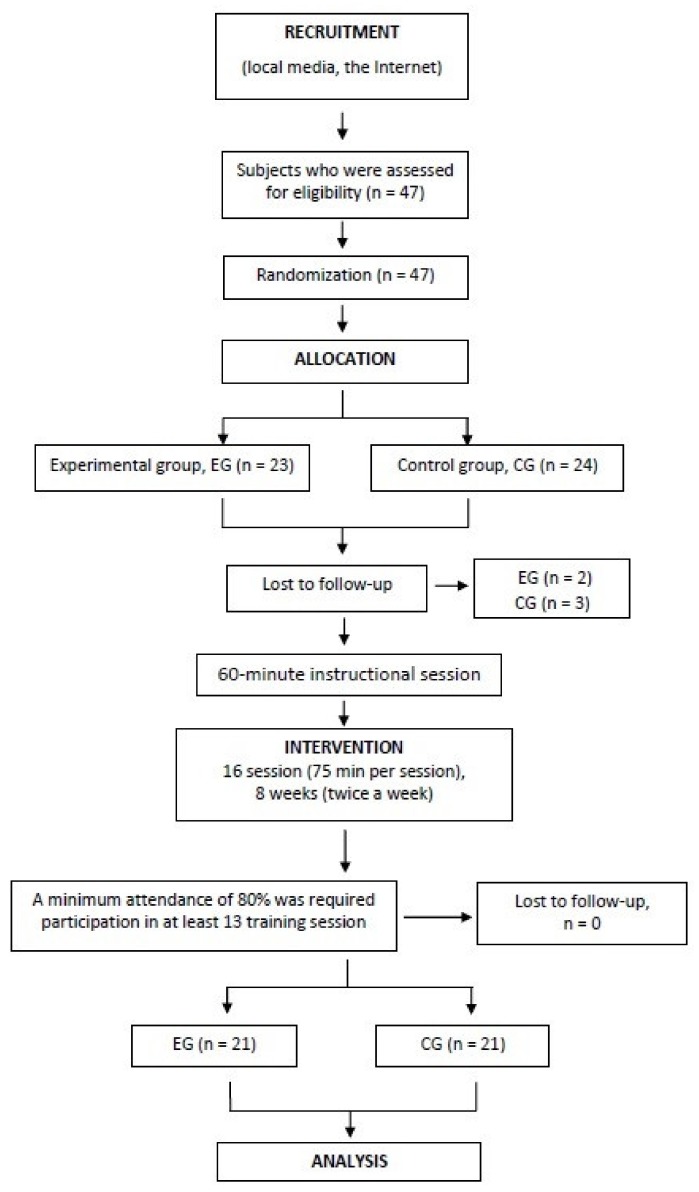
Flowchart of the study participants.

**Figure 2 ijerph-17-02197-f002:**
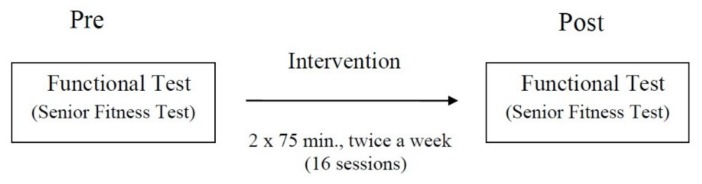
General overview of the experiment.

**Figure 3 ijerph-17-02197-f003:**
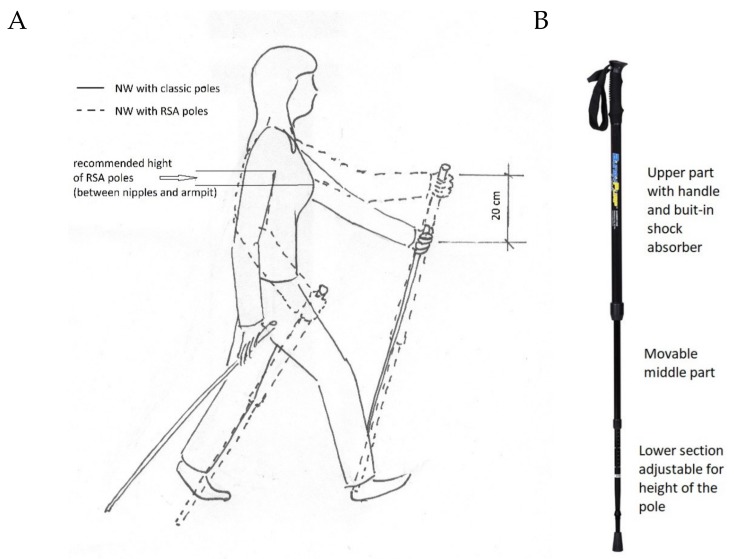
Correct position of upper limbs during Nordic walking with classic and resistant shock absorber (RSA) poles (**A**) and structure of RSA pole (**B**).

**Figure 4 ijerph-17-02197-f004:**
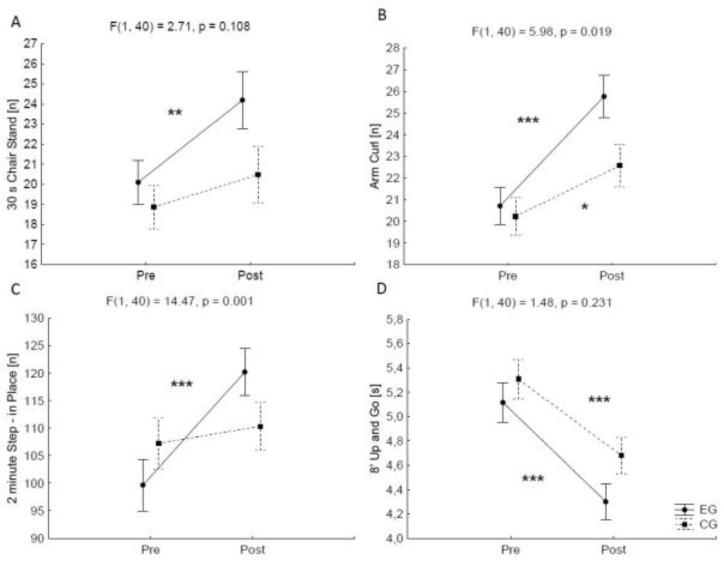
Mean values and standard error of measurements for 30-second chair stand, arm curl, 2-minute step in place and 8-foot up and go tests ((**A**–**D**), respectively) for the “time” factor (pre–post) in the experimental and control groups (EG and CG, respectively).* *p* < 0.05, ** *p* < 0.01, *** *p* < 0.001 (Scheffe post hoc test).

**Table 1 ijerph-17-02197-t001:** Average values and standard deviations for the general characteristics of the participants before the start of the experiment.

	EG (*n* = 21)		CG (*n* = 21)	
Measure	*M*	*SD*	*M*	*SD*
Age (years)	64.24	2.86	65.14	3.43
Body height (cm)	160.57	4.80	160.57	7.36
Body weight (kg)	71.90	11.52	66.57	9.89
BMI (kg/m^2^)	27.86	4.10	25.84	3.56

BMI—body mass index; EG—experimental group, CG—control group.

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
