# Peer review of "The Effect of Nordic Walking Training with Poles with an Integrated Resistance Shock Absorber on the Functional Fitness of Women over the Age of 60"

_ijerph, 2020, doi:10.3390/ijerph17072197_

Round 1

Reviewer 1 Report

This paper aims to evaluate the impact of Nordic walking training with poles for the functional fitness of older women. The author asked 42 women to do the experiment and analyzed the results.

However, the main problem of the paper is that the author did not provide the information of the method (no equations), and the experimental results are not enough. To improve the paper, please carefully refer to the paper with the title “The Effect of Treadmill Walking on Gait and Upper Trunk through Linear and Nonlinear Analysis Methods”. Please add related equations and more figures to illustrate the experiment and the results. And add figure to illustrate how to do the experiment clearly.

Author Response

We would like to thank Reviewer for detail suggestions.

We are very appreciate for suggesting the potential directions of interpretation of results as well as future directions of research. It’s a great idea to compare biomechanics of gait with poles (NW and RSA) and without poles. It would be useful in the explanation of the mechanisms of obtained results. Unfortunately, this direction of analysis in the study isn’t possible because it is finished now. However, taking into account the significance of  the problem we’ve inserted the following paragraph in discussion:

”In the analysis of the obtained results, it should be taken into account that characteristics of gait with the use of various NW poles (as well as without poles) might be different as in case of gait realized on a different surface. Shi et al. [33] observed differences in the structure of movement during overground walking and treadmill walking. They found shorter stride length, less stride time and a reduced correlation between gait and upper trunk features on a treadmill. A similar situation might occur in our study. It’s an open question if there is a difference in the kinematics of gait with or without NW poles as well as if applied training influenced on biomechanical characteristics of gait realized in everyday life. Shi et al. [33] suggested that changes in gait in non-standard conditions might be related to fall risk prevention which is very important among older people. Undoubtedly, it is an important direction for future studies.”  

Moreover, we’ve indicated the lack of biomechanical analysis as a limitation of the study.

On the other hand,  we would like to emphasize that this kind of analysis wasn’t the intention of our study. It’s a very important issue but the aim of our study was to check the effect of training on functional fitness among older women. Results are connected to simple/functional tests and were presented directly (in cm or seconds) rather than in the form of calculated indicators. That is why we didn’t insert any equations in the method section. However, according to suggestions,  we insert additional figures connected to the overall view of the experiment and the structure of poles.

Reviewer 2 Report

General comments:

This study investigated the impact of Nordic walking training with  integrated resistance shock absorber poles and  normal poles. The authors found both shock absorber and normal poles to be  beneficial for older women. It is clear what methodologies the authors used. Overall, the manuscript was well-written; however, there are several minor issues that need to be addressed.

Minor comments

  • A sentence on page 2, “physical activities are engaging for seniors and facilitate social contact.” sounds odd. The authors might want to rewrite it as follows, “seniors are engaging in physical activities that could promote social contact with others.”
  • The authors mentioned Nordic walking on page 2, but the term should be moved to the first or second paragraph on page 1.  
  • In 2.1 Participants, the authors should clarify what local media were used (such as local newspapers, TV shows, etc.) and say whether or not the Internet was used.  

Author Response

We would like to thank Reviewer for detail suggestions.

A sentence on page 2, “physical activities are engaging for seniors and facilitate social contact.” sounds odd. The authors might want to rewrite it as follows, “seniors are engaging in physical activities that could promote social contact with others.”

It has been changed taking into account also the other Reviewer’s suggestion:

“It is equally important that physical activities could promote social contact between people.”

The authors mentioned Nordic walking on page 2, but the term should be moved to the first or second paragraph on page 1. 

Following sentence has been added in the first paragraph:

“Nordic walking (NW) is one of the physical activities that can positively influence the walking ability.”

In 2.1 Participants, the authors should clarify what local media were used (such as local newspapers, TV shows, etc.) and say whether or not the Internet was used. 

It has been corrected

“The recruitment was carried out using ads in social media like the local newspapers and the Internet.”

Reviewer 3 Report

There are multiple places throughout the manuscript where a single sentence should be merged with others to form a complete paragraph.

Page 1

Merge the first sentence into the second paragraph. “Inadequate levels of physical activity among older people lead to a gradual decline in physical fitness, which can result in limited self-reliance and consequent dependence on other people [1-6].

Walking is one of the basic activities of everyday life.”

What does “Its quality…” mean? Are the authors referring to gait?

Why “natural sarcopenia”? Is there an unnatural? Remove natural.

Page 2

Rewrite the 1st sentence to read, “According to Sherrington et al. [15] a balance exercise program for seniors demonstrates an improved response compared to aerobic exercise.”

What is meant by engaging? Mentally engaged?

Nordic walking is popular in what geographical region?

Merge these two sentence, “Lee and Park [20] compared the effects of NW training with those of general exercises. The results confirmed a greater improvement in body balance and leg strength in the NW training group.”

What is meant by, “It has been reported…”? please replace with a citation.

Confusing sentence flow, “…marching with poles with an integrated resistance shock absorber (RSA).”

Rewrite this run-on sentence for improved clarity, “It is important to determine the effect of NW with RSA poles, taking into account that elements of resistance training are simultaneously performed with endurance activity, in contrast to combined training, in which endurance and resistance exercises are usually provided alternately as separate exercises.”

Page 4

Why is this information relevant? “There were 15 women with higher education and 6 women with secondary education in the EG. In the CG, there were 17 and 4 women, respectively.”?  If information is relevant move to the beginning of the section merging with the first paragraph.

Replace “basic features” with “subjects’ characteristics”

Merge the intervention first two sentences to read, “Both groups performed training sessions at the same time, twice a week for 8 weeks, for a total of 16 training sessions.”

Page 5

Rewrite this confusing sentence, “In both cases, the locomotive movement is compatible with the individual phases of the gait, and the movement of the articular joints is similar to that of a dynamic, free walking.” Additionally, is there a “static walking”?

Rewrite this confusing sentence, “The moment of elastic deformation of the shock absorber at the moment of release of the pressure is felt as a precipitation of the body from equilibrium.”

These reference points, “The correct height is set midway between the points defined by the auxiliary and breast” are a general area of the body and if a more specific area can be provided this would help individuals who have never gone NW. The locations could be added to Figure 2 and strengthen the manuscript.

Page 8.

Merge these two sentences, “Marching with RSA poles meets the above criteria. This is a workout that combines the strengths of an aerobic workout with resistance training.”

Add a citations to this, “According to the recommendations of the ACSM…”

Author Response

We would like to thank Reviewer for detail suggestions.

Page 1

Merge the first sentence into the second paragraph. “Inadequate levels of physical activity among older people lead to a gradual decline in physical fitness, which can result in limited self-reliance and consequent dependence on other people [1-6].

Walking is one of the basic activities of everyday life.”

The contents of the first sentence and the second sentence were joined.

“One of the basic activities of everyday life is walking which might be crucial in delaying decline in physical fitness and prevent limitation in  self-reliance and consequent dependence on other people [1-6].”

What does “Its quality…” mean? Are the authors referring to gait?

It has been corrected.

“Gait quality…”

Why “natural sarcopenia”? Is there an unnatural? Remove natural.

It has been corrected.

Page 4

Rewrite the 1st sentence to read, “According to Sherrington et al. [15] a balance exercise program for seniors demonstrates an improved response compared to aerobic exercise.”

It has been corrected.

What is meant by engaging? Mentally engaged?

It has been corrected.

“It is equally important that physical activities could promote social contact between people.”

Nordic walking is popular in what geographical region?

It has been corrected.

” NW is a popular type of physical activity among older people, especially in Europe.”

Merge these two sentence, “Lee and Park [20] compared the effects of NW training with those of general exercises. The results confirmed a greater improvement in body balance and leg strength in the NW training group.”

It has been corrected.

“The results of Lee and Park [20] confirmed a greater improvement in body balance and leg strength in the NW training group compared to the group doing general exercises.”

What is meant by, “It has been reported…”? please replace with a citation.

It has been corrected.

 Confusing sentence flow, “…marching with poles with an integrated resistance shock absorber (RSA).”

It has been corrected.

“The new form of NW is training with modified poles which allows combining aerobic and strength training.  These modified poles contain an integrated resistance shock absorber (RSA).”

Rewrite this run-on sentence for improved clarity, “It is important to determine the effect of NW with RSA poles, taking into account that elements of resistance training are simultaneously performed with endurance activity, in contrast to combined training, in which endurance and resistance exercises are usually provided alternately as separate exercises.”

It has been corrected.

Usually, in combined training endurance and resistance exercises are provided alternately. It is a question if the effect of training with RSA poles will be different, taking into account that elements of resistance training are simultaneously performed with endurance activity.”

Page 6

Why is this information relevant? “There were 15 women with higher education and 6 women with secondary education in the EG. In the CG, there were 17 and 4 women, respectively.”?  If information is relevant move to the beginning of the section merging with the first paragraph.

It has been removed.

 Page 8

Replace “basic features” with “subjects’ characteristics”

It has been corrected.

 Page 9

Merge the intervention first two sentences to read, “Both groups performed training sessions at the same time, twice a week for 8 weeks, for a total of 16 training sessions.”

It has been corrected.

“Both groups trained at the same time, twice a week for 8 weeks, for a total of 16 training sessions”

Page 10

Rewrite this confusing sentence, “In both cases, the locomotive movement is compatible with the individual phases of the gait, and the movement of the articular joints is similar to that of a dynamic, free walking.” Additionally, is there a “static walking”?

It has been corrected.

“In both cases, the body's movement is compatible with the individual phases of the gait, but limbs work with more power and in a wider range of motion than in free walking.”

Page 11

Rewrite this confusing sentence, “The moment of elastic deformation of the shock absorber at the moment of release of the pressure is felt as a precipitation of the body from equilibrium.”

It has been corrected.

“Release of the pressure from the RSA might cause a sensation of changes in body balance.”

These reference points, “The correct height is set midway between the points defined by the auxiliary and breast” are a general area of the body and if a more specific area can be provided this would help individuals who have never gone NW. The locations could be added to Figure 2 and strengthen the manuscript.

It has been corrected.

 “The correct height of the RSA poles is set midway between the points defined by the nipples and armpit”

Additional figure has been added to article.

Page 15

Merge these two sentences, “Marching with RSA poles meets the above criteria. This is a workout that combines the strengths of an aerobic workout with resistance training.”

It has been corrected.

“Marching with RSA poles combines elements of an aerobic workout with resistance training what is under ACSM recommendations.”

Add a citations to this, “According to the recommendations of the ACSM…”

It has been added.

Round 2

Reviewer 1 Report

The author has revised the manuscript according to the comment. It can be accepted.